# RABL6A Promotes Pancreatic Neuroendocrine Tumor Angiogenesis and Progression In Vivo

**DOI:** 10.3390/biomedicines9060633

**Published:** 2021-06-02

**Authors:** Chandra K. Maharjan, Shaikamjad Umesalma, Courtney A. Kaemmer, Viviane P. Muniz, Casey Bauchle, Sarah L. Mott, K. D. Zamba, Patrick Breheny, Mariah R. Leidinger, Benjamin W. Darbro, Samuel B. Stephens, David K. Meyerholz, Dawn E. Quelle

**Affiliations:** 1Department of Neuroscience and Pharmacology, The University of Iowa, Iowa City, IA 52242, USA; chandrakumar-maharjan@uiowa.edu (C.K.M.); umesalma-shaikamjad@uiowa.edu (S.U.); courtney-waite@uiowa.edu (C.A.K.); vpmuniz@gmail.com (V.P.M.); 2Fraternal Order of Eagles Diabetes Research Center, The University of Iowa, Iowa City, IA 52242, USA; cjb2cf@virginia.edu (C.B.); samuel-b-stephens@uiowa.edu (S.B.S.); 3Holden Comprehensive Cancer Center, The University of Iowa, Iowa City, IA 52242, USA; sarah-mott@uiowa.edu (S.L.M.); gideon-zamba@uiowa.edu (K.D.Z.); patrick-breheny@uiowa.edu (P.B.); benjamin-darbro@uiowa.edu (B.W.D.); 4Department of Biostatistics, The University of Iowa, Iowa City, IA 52242, USA; 5Department of Pathology, The University of Iowa, Iowa City, IA 52242, USA; mariah-leidinger@uiowa.edu (M.R.L.); david-meyerholz@uiowa.edu (D.K.M.); 6Department of Pediatrics, The University of Iowa, Iowa City, IA 52242, USA

**Keywords:** pancreatic neuroendocrine tumors, RIP-Tag2 mouse model, RABL6A, angiogenesis, *c-Myc*

## Abstract

Pancreatic neuroendocrine tumors (pNETs) are difficult-to-treat neoplasms whose incidence is rising. Greater understanding of pNET pathogenesis is needed to identify new biomarkers and targets for improved therapy. RABL6A, a novel oncogenic GTPase, is highly expressed in patient pNETs and required for pNET cell proliferation and survival in vitro. Here, we investigated the role of RABL6A in pNET progression in vivo using a well-established model of the disease. RIP-Tag2 (RT2) mice develop functional pNETs (insulinomas) due to SV40 large T-antigen expression in pancreatic islet β cells. RABL6A loss in RT2 mice significantly delayed pancreatic tumor formation, reduced tumor angiogenesis and mitoses, and extended survival. Those effects correlated with upregulation of anti-angiogenic p19ARF and downregulation of proangiogenic *c-Myc* in RABL6A-deficient islets and tumors. Our findings demonstrate that RABL6A is a bona fide oncogenic driver of pNET angiogenesis and development in vivo.

## 1. Introduction

Pancreatic neuroendocrine tumors (pNETs) are incurable, typically indolent neoplasms originating from multiple neuroendocrine cell types, particularly β cells, within the islets of Langerhans [1]. In healthy individuals, pancreatic β cells maintain proper blood glucose levels via insulin production and release. β cell dysfunction leads to diabetes while their neoplastic transformation results in pNET formation. Although pNETs are uncommon, their incidence has risen more than 4-fold over the past four decades [2,3,4]. This is alarming because the molecular etiology of pNET development is only partly understood and current therapies have little to no impact on improving overall patient survival [5]. Greater insight into essential mechanisms driving this disease will pave the way for more effective targeted therapies.

We recently identified a new potential driver of pNETs, a RAB-like GTPase named RABL6A. The protein was first discovered as a novel binding partner of the Alternative Reading Frame (ARF; p14ARF in humans, p19ARF in mice) tumor suppressor [6]. Later studies showed RABL6A is essential for the growth and survival of various tumor types, controls central cancer pathways (RB1, p53, PP2A, AKT, mTOR, ERK), and is a biomarker of poor survival in several cancers including breast and pancreatic adenocarcinomas [7,8,9,10,11,12,13,14,15]. In pNETs, RABL6A is highly expressed at the genetic and protein levels in human primary and metastatic tumors [10,16]. Functional studies in cultured pNETs revealed RABL6A is required for pNET cell proliferation and viability [10,12]. Moreover, RABL6A acts through several clinically relevant pNET pathways. This includes inhibition of the retinoblastoma (RB1) and protein phosphatases 2A (PP2A) tumor suppressors and activation of oncogenic AKT and mTOR [10,12]. Considered together, the in vitro analyses of RABL6A activity in cultured pNETs suggest it would promote pNET pathogenesis in vivo.

In this study, we explored the in vivo oncogenic role of RABL6A in pNET development and progression. We employed the RIP-Tag2 (RT2) transgenic mouse model, which has been used extensively to examine pNET biology and evaluate the utility of promising new therapies [17,18,19,20,21,22,23,24,25,26,27]. In RT2 mice, the rat insulin promoter (RIP) drives expression of the SV-40 large T-antigen in pancreatic β cells, causing their oncogenic transformation in a highly reliable, sequential, and age-dependent manner [21]. We found that loss of RABL6A in this pNET model significantly reduced the tumor phenotype and increased survival, demonstrating a critical in vivo role for RABL6A in pNET pathogenesis.

## 2. Materials and Methods

### 2.1. Mice

This mouse study tested the hypothesis that RABL6A loss by gene disruption will reduce pNET development in an established genetic model of the disease, RIP-Tag2 (abbreviated RT2). Animals were housed in barrier conditions. All mouse handling was conducted in strict compliance with The University of Iowa Institutional Animal Care and Use Committee (IACUC) policies (animal care protocol # 8111590), which adheres to the requirements of the National Institutes of Health Guide for the Care and Use of Laboratory Animals and the Public Health Service Policy on the Humane Care and Use of Laboratory Animals. All efforts were made to minimize animal suffering. Two RT2 male mice were kindly provided by Dr. Chris Harris (University of Rochester). Male RT2 mice (C57BL/6N strain) were bred with wild-type C57BL/6N females (Charles River) to obtain RT2 and WT experimental cohorts. RT2 males were mated with R6KO (i.e., Rabl6m/m, also on the C57BL/6N background) females to obtain RT2; R6+/m mice (50% incidence), which were further crossed with R6KO mice to obtain RT2; R6KO (i.e., RT2; Rabl6m/m) mice (25% incidence). Breeding pairs of RT2; R6KO males and R6KO females were used to generate R6KO and RT2; R6KO experimental cohorts at 50:50 ratio. Mice were grouped by genotype (i.e., WT, RT2, R6KO, or RT2; R6KO) and sex, and then randomly binned into the different experimental age groups (i.e., 8, 10, and 12 weeks for females, and 8, 12, and 16 weeks for males) for analyses. The final numbers ranged from 5–14 animals of each sex per genotype and time point for different assays as well as expanded numbers for survival analyses (which ranged from 23 to 45 mice per sex and genotype). Animals that died in pen (DIP) or those which were euthanized for having low BCS conditions were used in survival analysis.

### 2.2. Genotyping

All mice were genotyped by PCR. Genomic DNA were extracted from tail snips of the new weanlings using RED Extract-N-Amp™ PCR kit. Primers for *RT2* (+/tg) genotyping were obtained from NCI mouse repository: (a) Transgene: forward- ^5′^GGACAAACCACAACTAGAATGCAG^3′^, reverse- ^5′^CAGAGCAGAATTGTGGAGTGG^3′^, and (b) WT: forward- ^5′^CACCGGAGAATGGGAAGCCGAA^3′^, reverse- ^5′^TCCACACAGATGGAGCGTCCAG^3′^. *RT2* PCR cycling conditions: (a) 94 °C 3 min, (b) 35 cycles of 94 °C 30 s, 60 °C 30 s, 72 °C 30 s, and (c) 72 °C 3 min. *RABL6* genotyping was performed using a common forward primer- ^5′^CTACAGGACCTGTGGTTGTCT^3′^, and two reverse primers- ^5′^CTGGCTCTCATGGAATCGTG^3′^ (WT *RABL6* specific) and ^5′^CCAACTGACCTTGGGCAAGAACAT^3′^ (gene-trap mutant-specific). *RABL6* PCR cycling conditions: (a) 95 °C 10 min, (b) 35 cycles of 95 °C 15 s, 56.5 °C 45 s, 60 °C 1 min. PCR products were run on 1.5% agarose gel containing ethidium bromide and imaged using UVP BioSpectrum^®^ 610 Imaging System.

### 2.3. Mouse Blood Withdrawal and Insulin ELISA

Mice were starved for five hours prior to blood withdrawal. Their lateral saphenous vein was punctured using a 25G7/8 needle to collect 30–40 µL blood per mouse into a heparinized capillary tube. The blood was immediately mixed with 1 μL of 5% EDTA to prevent clotting and later centrifuged at 5000 rcf for 5 min at 4 °C. The supernatant (plasma) was carefully collected using a pipette and stored at −80 °C. To measure plasma insulin levels, frozen plasma was thawed on ice and 5 μL/sample loaded in duplicate wells onto a 96 well plate in Mouse Ultrasensitive Insulin ELISA kit from ALPCO^®^. Absorbance of the final products was measured at 450 nm using BioTek^®^ Synergy 4 plate reader. Data analysis was performed using GraphPad Software.

### 2.4. Islet Isolation

Mice were euthanized by cervical dislocation following isoflurane-induced anesthesia in a glass chamber. Pancreatic islets of Langerhans (including tumorigenic islets) were isolated using pancreatic perfusion technique [28]. A total of 2–3 mL of 0.8 mg/mL collagenase in HBSS solution was perfused into the pancreas through the cannulated common bile duct (clamped at its hepatic branch). The inflated pancreas was carefully removed and digested at 37 °C for 10 min. Islets and tumors were then released by gentle agitation, washed in RPMI with 1% FBS and purified on Histopaque 1077 and 1119 gradients. Islet pellets and tumors were flash frozen in liquid nitrogen and stored at −80 °C.

### 2.5. RNA Isolation and qRT-PCR Analysis

RNA was prepared from isolated islets using Qiagen RNAeasy^®^ Plus Mini Kit. 40 µL of 1M dithiothreitol (DTT) was added per mL of lysis buffer to inhibit abundantly expressed pancreatic ribonucleases [29]. cDNA was synthesized from 100–200 ng RNA using SuperScript^®^ III First Strand cDNA preparation kit (Invitrogen). Diluted cDNA was then used for qPCR (cycling conditions: denaturation at 95 °C for 10 min followed by 40 cycles of 95 °C for 15 s and 60 °C for 1 min) with gene-specific primers in the presence of iQ^TM^ SYBR^®^ Green Supermix reagent on a Bio-Rad CFX96™ Real-Time System. Fold changes in each gene mRNA levels were calibrated to *Hprt* mRNA expression and computed using the 2^-∆∆Ct^ method.

### 2.6. qRT-PCR Primers

The National Center of Biotechnology Information (NCBI) Primer-BLAST online tool was used to design primers specific for the mouse genes of interest. (a) *Hprt* (housekeeping gene used for normalization): forward- ^5′^GCCCCAAAATGGTTAAGGTTG^3′^, reverse- ^5′^TGGCCTGTATCCAACACTTC^3′^; (b) *c-Myc*: forward- ^5′^CCTGTACCTCGTCCGATTCC^3′^, reverse- ^5′^TTCTTGCTCTTCTTCAGAGTCG^3′^; (c) *Vegfa*: forward- ^5′^AGCAGATGTGAATGCAGACCA^3′^, reverse- ^5′^GACCCAAAGTGCTCCTCGAA^3′^; and (d) *Arf*- forward- ^5′^CGGAATCCTGGACCAGGTG^3′^, reverse- ^5′^ACCAGCGTGTCCAGGAAGC ^3′^.

### 2.7. Analysis of Tumor Burden

Tumors (lesions ≥ 1 mm) were excised from a fully perfused pancreas and their dimensions measured with a ruler. Individual tumor volumes were calculated using the formula: volume = width^2^ × length × 0.52 and summed up to calculate tumor burden per mouse as previously described [27]. Tumors were flash frozen in liquid nitrogen for western blot analysis.

### 2.8. Histopathological Analyses

Isolated pancreas tissues were fixed in formalin (10% neutral buffered formalin), paraffin-embedded, and processed for hematoxylin and eosin (H&E) staining. Histopathological scoring of tissues was performed in the masked manner and by following principles and approaches of reproducible scoring analysis [30]. Endocrine area, angiogenic islets, vasculature, and number of mitoses were analyzed on H&E-stained specimens using a brightfield microscope. To compute percentage endocrine area, the area covered by islets (normal and tumorigenic) in three randomly selected pancreatic regions viewed at low (20×) magnification was divided by the total area (% endocrine area = islet area/total pancreatic area ×100). The percentage of angiogenic islets was calculated as the number of angiogenic islets divided by the total (non-angiogenic plus angiogenic) multiplied by 100. To calculate the percent vasculature (i.e., vessels larger than capillaries), individual angiogenic islets were observed at 100× magnification and the ratio of area covered by blood vessel to the total area of an islet multiplied by 100 (% vasculature = vessel area/islet area ×100). Number of mitoses were counted on 3–4 random fields of view within an islet at 600× magnification. Percentage vasculature and number of mitoses were quantified on 3–4 randomly selected enlarged islets in the pancreas of RT2 and RT2; R6KO mice.

### 2.9. Western Blotting and Antibodies

Flash frozen tumors were stored at −80 °C and pulverized in LN2 using a mortar and pestle. Samples were lysed for 30min on ice with RIPA buffer (50 mM Tris, pH 8.0, 150 mM NaCl, 1% Triton X-100, 0.1% SDS, 0.5% sodium deoxycholate) containing 1 mM NaF, protease and phosphatase inhibitor cocktails (Sigma, P-8340 & P-0044) as well as 30 µM phenylmethylsulfonyl fluoride (PMSF). Protein concentrations of the extracts were determined by BCA protein assay (ThermoFisher, Cat. # 23228). Equivalent amounts of protein per sample were separated by SDS-PAGE and transferred onto PVDF membranes (Millipore). Membranes were blocked with 5% non-fat milk or 5% BSA in TBST (Tris-buffered saline containing Tween-20) depending on the antibody. Proteins were detected using HRP-conjugated secondary antibodies and enhanced chemiluminescence (ECL, Amersham, Buckinghamshire, UK). Densitometry quantification was performed using ImageJ (NIH). Antibodies were used according to supplier guidelines and included: GAPDH mouse monoclonal (no. Ab8245) from Abcam; Myc [Y69] rabbit monoclonal (no. ab32072) from Abcam; VEGFA [VG-1] mouse monoclonal (no. sc-53462), Chromogranin A rabbit polyclonal (no. ab45179); p19ARF rabbit polyclonal (no. NB200-106); HRP-coupled secondary antibodies (nos. NA934 and NA935) from Sigma; and RABL6A polyclonal rabbit antibody produced in the Quelle laboratory [6,8].

### 2.10. RNA Interference in pNET Cells

RABL6A was silenced in human BON-1 pNET cells using pLKO.1.puro based lentiviral constructs containing two different shRNAs to RABL6A, as described previously [8,10,12]. Briefly, lentiviruses encoding RABL6A shRNAs (KD1 and KD2) and empty vector control were generated in HEK293T cells following lipofection of the viral pLKO.1 and helper constructs. BON-1 cells were infected twice with virus supernatants and harvested 5 days after infection according to established methods [10,12]. BON-1 cells (provided by Dr. Courtney Townsend, University of Texas Medical Branch, Galveston, TX, USA) [31] were grown in Dulbecco’s modified Eagle’s medium (DMEM)/F12 containing 10% fetal bovine serum (FBS), 4 mM glutamine, and 100 µg/mL penicillin-streptomycin. Human embryonic kidney (HEK) 293T cells were maintained in DMEM, 10% FBS, 4 mM glutamine and 100 µg/mL penicillin-streptomycin. Cells were routinely tested for mycoplasma contamination and found to be negative.

### 2.11. Statistics

Western data were imaged by scanning densitometry and quantified by ImageJ (NIH). Quantified data were presented as the mean +/− SEM for the biological replicates in each analysis. Linear regression was utilized to evaluate differences in tumor volume, average number of number of mitoses, transcript levels, and protein levels; a log-transformation was applied to each outcome prior to analysis to meet model assumptions. Beta regression models were utilized to evaluated differences in the proportion of endocrine area, angiogenic islets, and vascular area per islet. The Kaplan–Meier method was used to estimate the survival curves, and group comparisons were made using the log rank test. Cox regression was used to evaluate the overall effect of genotype while adjusting for sex. Linear mixed effects regression models were used to estimate and compare group-specific changes in plasma insulin over time; a log-transformation was applied to stabilize the variance. Accounting for unequal variances or random variability attributable to repeat measurements within a mouse was done, as necessary. All tests were two-sided and assessed at the 5% significance level using SAS v9.4 (SAS Institute, Cary, NC, USA).

## 3. Results

### 3.1. Description and Generation of the Mouse Models

RT2 mice exhibit temporally defined, multi-stage development of pNETs through oncogenic transformation of the islet β cells [21]. Over half of the normal pancreatic islets in RT2 mice proliferate aberrantly and transform into hyperplastic islets by 3–5 weeks. Of the hyperplastic islets, about one-fifth undergo an angiogenic switch and become vascularized angiogenic islets by 7–9 weeks, with one-fourth of those progressing into enlarged pNETs starting at 10–12 weeks [20,21,32]. The pNETs that arise are functional insulinomas that secrete insulin. Representative hyperplastic islets, red angiogenic islets, and red vascularized pNETs isolated by pancreatic perfusion from 12-week-old RT2 male mice are shown in Figure 1A.

To examine the oncogenic role of RABL6A in pNET development, we first generated *Rabl6* knockout (R6KO) mice. *Rabl6* chimeric mice were obtained from the Knockout Mouse Project (KOMP) and bred with C57BL/6N animals to achieve germline transmission. A gene trap sequence consisting of a splice acceptor and a poly(A) adenylation site was targeted between 5′ exons 3 and 4 of the *Rabl6* gene (Figure 1B), interfering with normal gene splicing and expression. After successful germline transmission, the heterozygous *Rabl6^+/m^* mutant mice were interbred to obtain homozygous *Rabl6^m/m^* mice, henceforth referred to as RABL6A knockout (R6KO). A classical 1:2:1 Mendelian ratio was obtained from intercrosses of *Rabl6^+/m^* mice demonstrating that RABL6A deficiency is not embryonically lethal (Figure 1C). Western analysis verified loss of RABL6A protein expression in homozygous R6KO pancreatic and brain lysates (Figure 1D).

RT2 and R6KO mice were crossed for multiple generations to obtain double transgenic RT2; R6KO mice as well as RT2, R6KO, and wild-type (WT) controls (Figure 1E; genotyping results shown in Appendix A).

### 3.2. RABL6A Expression Reduces Survival in RT2 Mice

RT2 mice have a short life expectancy due to excessive insulin secretion from the pNETs that induces hypoglycemic shock and death [21]. We monitored the survival of male and female mice within all four experimental cohorts to determine if RABL6A loss improves survival in RT2 mice. All WT and R6KO mice of both sexes were alive throughout the entire observation period. A significant difference was seen in the survival of RT2 females (median 13.3 weeks) versus RT2 males (median 15.7 weeks) (Figure 2), an observation that we could not find noted in prior reports where the sex of the animals was often not stated. As hypothesized, loss of RABL6A significantly increased the survival of RT2; R6KO males (median 18.0 weeks). A modest increase in the median survival of RT2; R6KO females (14.1 weeks) was seen relative to that of RT2 females (13.3 weeks), but the difference was not statistically significant (Figure 2; Table 1). After adjusting for sex, an overall difference in survival was evidenced between RT2 and RT2; R6KO mice with RT2 mice being at 59% increased risk of death (HR = 1.59, *p* < 0.05).

### 3.3. RABL6A Increases Tumor Burden in RT2 Mice in an Age-Dependent Manner

Given the sex-dependent differences in overall survival for RT2 and RT2; R6KO mice, we evaluated the development and progression of tumors according to different timelines for male and female animals. Female cohorts were euthanized for histopathologic and molecular analysis at 8, 10, and 12 weeks of age whereas males were euthanized at 8, 12, and 16 weeks of age (Figure 3A). At each time point, pancreatic islets and tumors were isolated via pancreatic perfusion or the whole pancreas was excised for fixation and histopathology.

We first quantified the volume of tumors harvested from RT2 and RT2; R6KO mice and observed an age-dependent decrease in tumor burden in mice lacking RABL6A (Figure 3B; Table 1). Specifically, 10-week-old female and 12-week-old male RT2; R6KO mice had significantly lower tumor burden compared to their RT2 counterparts. The difference in tumor burden associated with RABL6A loss was absent in older animals, reflecting a delay rather than abolishment of tumor formation (Figure 3B). Since pNETs in RT2 mice cause hyperinsulinemia, we tested whether plasma insulin levels correlated with tumor burden (Appendix A). WT and R6KO mice, which both lack tumors, exhibited normal, low plasma insulin concentrations (<2 ng/mL) throughout the observation timeframe. Notably, R6KO male mice had reduced insulin levels compared to WT mice as the animals aged. By comparison, plasma insulin levels increased at similar rates in both RT2 and RT2; R6KO mice (males and females) over time regardless of differences in tumor burden.

Complementary histopathological analyses of H&E-stained pancreata from each experimental cohort enabled further assessment of tumor burden. WT and R6KO mice had small, normal looking islets (Figure 3C) that represented a small fraction of the entire pancreas, as measured by the percent endocrine area (cumulative area of islets relative to the total pancreatic area) (Figure 3D). The pancreas of RT2 males and females contained enlarged islets (comprising hyperplastic islets, angiogenic islets, and tumors), which were invasive on the surrounding exocrine tissue (Figure 3C). Quantitative analysis revealed remarkably high percentage endocrine area in RT2 mice compared to WT and R6KO animals. The 8- and 10-week-old RT2; R6KO females had smaller islets and significantly reduced percent endocrine area compared to age-matched RT2 counterparts (Figure 3C,D; Table 1; Appendix A). A lower average percent endocrine area was likewise observed in 12-week-old RT2; R6KO males (13.5%) compared to RT2 males (17.7%), albeit smaller in magnitude and not statistically significant (Figure 3D; Table 1; Appendix A).

### 3.4. RABL6A Promotes Angiogenesis during Early Stages of pNET Formation

In RT2 mice, the progression of hyperplastic islets to angiogenic islets, referred to as the angiogenic switch, is a key event in pNET development [22]. p19^ARF^, a well-known tumor suppressor and binding partner of RABL6A, was previously shown to disrupt the angiogenic switch and tumor initiation in RT2 mice [27]. To determine if RABL6A promotes pNET angiogenesis in RT2 mice, we compared the percentage of angiogenic islets and their vasculature in H&E-stained RT2 and RT2; R6KO pancreatic sections (Figure 4).

Angiogenic islets, characterized by pockets of blood vessels (capillaries not counted) contained within their circumference, were absent in control WT and R6KO pancreas (Figure 4A,B). In RT2 mice, the number of angiogenic islets relative to the total islets increased with age (Figure 4A, Appendix A). In female mice, 10-week-old RT2 females displayed a significantly higher percentage of angiogenic islets compared to their RT2; R6KO counterparts (Figure 4A,B; Table 1; Appendix A). While the differences between 12-week-old male cohorts did not reach statistical significance, the average percentage of angiogenic islets trended towards being similarly greater in RT2 mice relative to RT2; R6KO animals (31.4% to 21.9%) (Figure 4A; Table 1).

We then asked whether RABL6A deficiency attenuates the extent of vasculature within angiogenic islets in RT2 mice. For this, we quantified percentage area of vasculature in 3–6 representative angiogenic islets in each RT2 or RT2; R6KO pancreas. RT2 angiogenic islets were highly vascularized and the extent of vasculature was significantly reduced in the angiogenic islets of 10-week-old RT2; RABLKO females (Figure 4B,C; Table 1). This effect was not observed in RT2; R6KO males compared to age matched RT2 males at 8, 12, or 16 weeks although a similar downward trend in angiogenic islets was observed in 12-week RT2; R6KO males (Figure 4C; Table 1; Appendix A). These cumulative findings demonstrate that RABL6A promotes the angiogenic switch and increases vascularization during the early stages of pNET pathogenesis in RT2 females with a less pronounced, non-statistically significant effect observed in RT2 males.

### 3.5. RABL6A Drives pNET Cell Mitosis Early in Tumor Development

RABL6A promotes pNET cell proliferation in vitro [10,12]. Mitotic counts on H&E sections are one of the grading parameters for pNETs [33]. Thus, we scored the number of mitoses (cells with condensed nuclear DNA) in non-overlapping high-power fields of pancreatic islets and tumors in RT2 and RT2; R6KO animals (Figure 5; Appendix A). A significant reduction in mitoses was seen in both 8- and 10-week-old female RT2; R6KO islets compared to those in age-matched RT2 mice (Figure 5A,B; Table 1; Appendix A). No significant differences in mitotic counts were seen in the islets and pNETs of older (12-week) female RT2; R6KO and RT2 mice or in 8-week-old male cohorts (Appendix A). However, 12-week-old RT2; R6KO males displayed a modest decrease in mitoses (mean 2.4 vs. 3.0 in RT2 controls; not significant) that mirrored the differences in younger female cohorts (Figure 5A,B; Table 1; Appendix A). Of note, 16-week-old RT2; R6KO males showed elevated pNET mitoses relative to age-matched RT2 controls (Appendix A), possibly reflecting compensatory hyperactivation of proliferative signaling following sustained loss of RABL6A [12]. Together, these results suggest RABL6A enhances islet cell mitosis at early time points in PNET development, with effects most evident in female mice.

### 3.6. RABL6A Promotes c-Myc and Limits ARF Expression in Mouse Pancreatic Islets

Targeted overexpression of ectopic *c-Myc* in pancreatic β cells yields highly angiogenic islet tumors in mice while its deactivation induces tumor regression preceded by vascular degeneration and β cell apoptosis [34]. Other work in RT2 mice showed that endogenous *c-Myc* activity is required for maintaining pNETs and their vasculature [26]. Prior microarray analyses of RABL6A-depleted human pNET cell lines showed significant reduction in *c-Myc* mRNA levels [10], suggesting that RABL6A may normally promote *c-Myc* transcription.

To determine if RABL6A promotes the expression of *c-Myc* mRNA in pancreatic islets, quantitative RT-PCR assays were performed on cDNA from the isolated islets of WT vs. R6KO and RT2 vs. RT2; R6KO mice. Significant downregulation of *c-Myc* expression was seen in normal islets from 8–10-week-old female and 12-week-old male R6KO mice lacking RABL6A compared to age-matched WT controls (Figure 6A). To assess the effect of RABL6A loss on *c-Myc* levels in neoplastic lesions, identical analyses were performed using transformed islets from RT2 and RT2; R6KO pancreata. The pancreatic perfusions from these animals yielded a mixture of normal, hyperplastic, and angiogenic islets. As shown in Figure 6B and Table 1, *c-Myc* mRNA expression was significantly reduced by ~2-fold in neoplastic islets from 8–10-week RT2; R6KO females vs. RT2 controls (*p* < 0.05). RABL6A deficiency in RT2; R6KO 12-week males likewise yielded a more than 3-fold average decrease (albeit not statistically significant, *p* = 0.11) in *c-Myc* expression (Figure 6B; Table 1). These data suggest RABL6A promotes *c-Myc* mRNA expression in normal and transformed pancreatic islets.

Western analyses of isolated pNETs from RT2 versus RT2; R6KO mice were performed to determine if RABL6A deficiency caused reduced expression of *c-Myc* protein (Figure 6C; Table 1). Pancreatic tumors from 10-week female cohorts were either absent or too small to be examined by western blotting; consequently, tumors from 12-week-old male mice were assayed. Immunoblotting for RABL6A verified its absence in RT2; R6KO tumors whereas high levels of expression were seen in RT2 lesions. Although results did not reach statistical significance, the majority of RT2; R6KO tumors expressed lower levels of *c-Myc* protein compared to RABL6A-positive RT2 tumors (Figure 6C,D; Table 1).

VEGFA is a known pro-angiogenic gene in pNETs that is transactivated by *c-Myc* [18,23,35]. qRT-PCR and western blot analyses were performed to test whether RABL6A upregulates VEGFA expression in WT-normal and RT2-transformed islets. We found that RABL6A promotes *Vegfa* transcription in the normal islets of WT male and female mice, but this was not seen in the transformed islets of RT2 mice (Appendix A). Likewise, western analyses of isolated pNETs showed no significant difference in VEGFA protein levels between 12-week-old RT2 and RT2; R6KO males although a modest decrease in the mean value was observed in the RABL6A-deficient tumors (Figure 6C; Table 1). Immunoblotting for the neuroendocrine specific differentiation marker, chromogranin A (CgA), verified retention of pNET identity in tumors lacking RABL6A (Figure 6C,D).

Prior studies in RT2 mice demonstrated an important role for the p19ARF tumor suppressor in reducing pNET development [27]. Specifically, genetic deletion of *Arf* significantly accelerated tumor formation by promoting the angiogenic switch. Since RABL6A promotes pNET angiogenesis and physically associates with the ARF protein [6], we speculated RABL6A may impair ARF expression. qRT-PCR analyses of *Arf* transcript levels in pancreatic islets from WT vs. R6KO and RT2 vs. RT2; R6KO mice suggest that RABL6A does not regulate p19ARF expression at the transcriptional level (Appendix A). To test whether RABL6A negatively regulates p19ARF protein expression, the same tumor samples from Figure 6C (top panels) were examined by western blotting for mouse p19ARF (bottom panels, minus tumor #7 because sample was used up). As predicted, loss of RABL6A in RT2; R6KO tumors correlated with a significant, 4.2-fold average increase in p19ARF expression (Figure 6C,D; Table 1). These exciting findings support a model in which oncogenic RABL6A may promote pNET angiogenesis and progression through upregulation of *c-Myc* and/or downregulation of ARF in pancreatic islets (Figure 6E).

To assess the RABL6A-Myc relationship in human pNET cells, we silenced RABL6A in BON-1 cells using two different, validated shRNAs [8]. Loss of RABL6A was confirmed by western blotting and correlated with significant (5- to 10-fold) downregulation of *c-Myc* protein (Figure 6F,G). RABL6A knockdown also caused a marked decrease in cell survival (Figure 6F), as previously reported in these cells [10,12]. A more modest (~2-fold) reduction in *c-Myc* mRNA expression was observed in RABL6A-depleted cells (Figure 6H). These data demonstrate that RABL6A normally promotes *c-Myc* expression, likely through transcriptional and post-transcriptional mechanisms in pNET cells.

## 4. Discussion

Molecular alterations driving pNET pathogenesis are only partly understood, but what is known has helped guide current patient therapies. Standard treatments such as everolimus (inhibitor of mammalian target of rapamycin [mTOR]) and sunitinib (inhibitor of receptor tyrosine kinases, including VEGFR) [36,37,38] were supported by patient tumor profiling [39,40] and drug studies in RT2 mice demonstrating their effectiveness [19,24]. However, none of the current therapies for pNETs are curative and overall patient survival is not significantly improved. Greater insight into key drivers of the disease is still needed to inform new treatments with innovative targeted and/or combination therapies. This study explored the in vivo significance of a relatively new player in pNET biology, RABL6A. We found that loss of RABL6A in the RT2 pNET model reduced tumor burden, angiogenesis, and mitoses in an age-dependent manner that coincided with improved survival.

Several prior observations predicted that RABL6A would play an oncogenic role in pNET pathogenesis in vivo. First, we have previously established that RABL6A is upregulated at the genetic and protein level in patient pNETs [10,16]. Second, RABL6A promotes pNET cell survival and proliferation through multiple, clinically druggable pathways that include CDK4/6-RB1, PP2A, and Akt/mTOR [10,12]. Third, microarray analyses of cultured pNET cells expressing or lacking RABL6A identified a RABL6A-regulated gene expression profile that included activation of *c-Myc*, VEGFR, and EGFR pathways [10], all of which are major mediators of tumor angiogenesis.

Angiogenesis is critical to pNET development and progression. This is evidenced by the clinical use of sunitinib to block pro-angiogenic receptor tyrosine kinases, like VEGFR, and their effectors (PI3K/Akt/mTOR, PKCs, Ras/MAPK) in pNET therapy [36,41]. Oncogenic *c-Myc*, a downstream target of Akt/mTOR signaling [42,43], transcribes VEGF [35], a VEGFR ligand, and is critical for pNET angiogenesis and progression [18,23,26,34]. Our data revealed no correlation between VEGFA and RABL6A expression in transformed islets and tumors, whereas *c-Myc* transcripts were downregulated in RABL6A-deficient pancreatic islets, in keeping with reduced angiogenic islets and vasculature in young RT2; R6KO mice. Those results support the conclusion that RABL6A may promote pNET angiogenesis in RT2 mice, at least in part, via *c-Myc* upregulation (as depicted in Figure 6E). Consistent with that notion, *c-Myc* protein was markedly downregulated in 7 of 8 RT2 tumors lacking RABL6A. *c-Myc* has many transcriptional targets besides VEGF that may mediate its angiogenic effects, including activation of HIF-1α [44], repression of Thrombospondin-1 [45,46], and induction of interleukin-1β [47]. How RABL6A affects the expression and activity of these *c-Myc* targets is an important line of future investigation.

Other RABL6A effectors likely contribute to its pro-angiogenic activity in pNET development. RABL6A promotes AKT-mTOR signaling via inactivation of the PP2A tumor suppressor [12], and numerous studies have shown that activated AKT-mTOR signaling promotes tumor angiogenesis [48]. The role of PP2A in angiogenesis is less clear but a recent study of a vascular neoplasm, called hemangioma, revealed that inactivation of PP2A promotes angiogenesis and hemangioma formation [49]. Notably, RABL6A physically interacts with the ARF tumor suppressor [6]. Since ARF suppresses pNETs in RT2 mice by blocking the angiogenic switch [27], RABL6A may promote tumor angiogenesis by antagonizing ARF. That concurs with our finding that p19ARF protein expression is significantly reduced in RABL6A-expressing pNETs. It is possible that RABL6A, which is predominantly cytosolic but shuttles into the nucleus [6], mobilizes ARF from the nucleus and nucleoli (where it is most stable) into the cytosol where it undergoes proteasomal and lysosomal degradation [50,51,52,53,54]. Further investigations in cells and mice with altered RABL6A and ARF will be needed to test these possibilities.

The RT2 mouse model is broadly used in pNET research. It provides a reliable, fully penetrant model of multi-stage pNET progression that has been validated for preclinical drug testing. Besides demonstrating in vivo anti-tumor activity of rapamycin and sunitinib [19,24], RT2 mice have helped rule out pathways with low clinical relevance in pNETs, such as IGF-1R [55,56]. Nevertheless, there are some limitations of this model. Inhibition of central tumor suppressors, p53 and RB1, by SV-40 large T-antigen in RT2 mice causes aggressive, highly proliferative pNETs. In this regard, the RT2 molecular signatures and phenotype more closely recapitulate high grade 3 (G3) insulinomas, which are well-differentiated pNETs that have a high proliferative index (Ki-67 > 20%). Moreover, the absence of functional RB1 likely reduced the extent to which RABL6A loss suppressed pNET formation since RABL6A functions partly through RB1 inhibition [10]. On the other hand, many low G1/G2 pNETs lack functional RB1 due to overexpression of CDK4 and CDK6 or loss of CDK inhibitors, like p16INK4a [57,58,59]. Thus, the RT2 model enabled us to determine the contribution of RABL6A to pNET development in a clinically relevant, RB1 inactive setting common in patient tumors. Notably, earlier drug studies in RT2 mice targeting *c-Myc* [26], VEGFA [17,18], and PI3K/Akt/mTOR signaling [60], yielded similar phenotypes of reduced tumor progression and angiogenesis as seen in RT2; R6KO animals.

It will be important to explore the in vivo role of RABL6A in pNET pathogenesis using other relevant models of the disease. Several groups have modeled deficiency of the multiple endocrine neoplasia 1 (*Men1*) gene in mice using the rat insulin promoter to achieve inactivation in β cells [61,62]. Patients with the MEN1 familial tumor syndrome develop multiple endocrine tumors, such as insulinomas, and somatic mutations of human *MEN1* are also common in various sporadic NETs, including those of the pancreas [39,63,64]. In mice, loss of *Men1* alone yields insulinomas with high penetrance but long latency period (80–100% incidence by ~1 year of age) [61,62]. The tumors develop in a multi-stage fashion similar to RT2 mice, with early formation of atypical hyperplastic islets progressing into angiogenic islets and ultimately low-grade neoplastic lesions, with no evidence of metastasis. Others recently demonstrated greatly accelerated formation of low-grade (G1/G2) insulinomas in mice with targeted inactivation of both *Men1* and *Pten*, with an onset of pNETs at 7 weeks of age [65]. If RABL6A promotes tumor angiogenesis and development in these contexts, as expected, its loss would diminish the tumor phenotype.

Although both male and female mice showed delayed tumor formation due to RABL6A loss, more pronounced anti-tumoral effects of RABL6A loss were generally seen in female RT2 mice. At face value, this might suggest potential sex-dependent activities of RABL6A, but several observations argue against that conclusion. First, the increase in survival associated with RABL6A deficiency was more significant in RT2 males although females trended in the same direction. Indeed, biostatistical analyses of all animals established significantly increased survival, regardless of sex, in mice lacking RABL6A. Second, females and males were analyzed at different time points, which were chosen because of the shorter survival of RT2 females. Consequently, analyses in males were further spread out (8, 12, and 16 weeks) than in females (8, 10, and 12 weeks) and lacked the 10-week time point where the most significant changes were observed in RT2; R6KO females. By 12 weeks of age, female RT2; R6KO mice no longer displayed the significant reductions in tumor burden, angiogenesis, or islet cell proliferation seen at earlier ages (mostly at 10 weeks). Those results establish an age-dependent effect of RABL6A loss on RT2 tumor phenotype. As such, it seems most likely that the timing of our analyses, not sex, impacted our ability to detect effects of RABL6A loss on the tumor phenotype in males.

Interestingly, the oldest mice examined in our study (16-week-old RT2; R6KO males) displayed increased mitoses compared to their RT2 counterparts and a trend toward greater tumor burden. This may reflect the development of resistance to the anti-tumor effects of sustained RABL6A loss, ultimately enabling tumors to progress more aggressively. This mirrors an unwanted problem (drug resistance) that invariably arises in the clinic during anti-cancer therapy, which has been observed in prior RT2 drug studies. For example, treatment of RT2 mice with anti-angiogenic drugs, sunitinib or VEGFR2 inhibitor, causes temporary tumor regression followed by evasive drug resistance marked by increased invasiveness and malignancy. The resistance against anti-angiogenic therapy was mediated by upregulation of pro-angiogenic factors, FGF2 [18] and c-Met [66,67]. Similar mechanisms may have enabled RT2; R6KO tumors to circumvent the reduced *c-Myc* and upregulation of p19ARF caused by RABL6A loss, possibly facilitating the progression of tumors in older male mice.

In sum, this work demonstrates a significant role for RABL6A in driving pNET angiogenesis and development in vivo. RABL6A oncogenic activity was coincident with increased expression of *c-Myc* and reduced ARF in pancreatic islets and tumors. Together, these results highlight the potential benefit of pharmacologically targeting multiple RABL6A pathways in pNET therapy and warrant further investigation into mechanisms of RABL6A action in pNET pathogenesis.

## Figures and Tables

**Figure 1 biomedicines-09-00633-f001:**
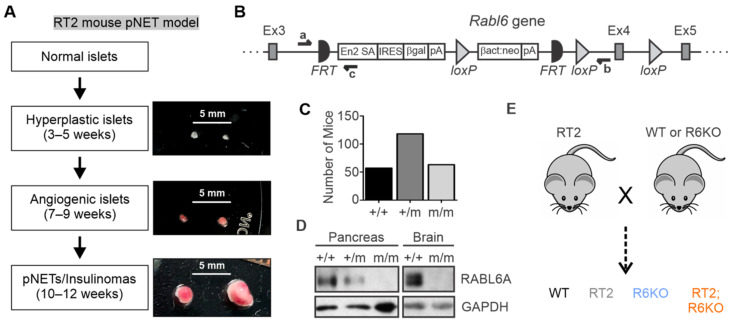
Description of the RT2 insulinoma and *Rabl6* deficient mouse models. (**A**) Schematic outlining the sequential progression of islet transformation into functional pNETs (insulinomas) in RT2 mice. Time in parentheses represent the age of RT2 mice when the indicated islets are first observed. Right: Images of each type of lesion (with scale bars) isolated from a 12-week-old RT2 male. (**B**) Diagram of the targeted *Rabl6* gene locus (exons 3–5 shown) in the gene-trap knock-out mouse model. Indicated sites include FRT (Flippase recombinase), loxP (Cre-recombinase), and primers for detecting endogenous *Rabl6* allele (‘a’ and ‘b’) and mutant (m) allele (‘a’ and ‘c’). (**C**) Number of mice with +/+, +/m, or m/m genotype from *Rabl6+/m* heterozygous mouse crossings showing a Mendelian distribution of 1:2:1. (**D**) Western blots showing loss of RABL6A protein in *Rabl6m/m* mouse pancreas and brain. (**E**) Simplified schematic of RT2 crosses with WT or *Rabl6m/m* (R6KO) mice over multiple generations to obtain the experimental cohorts (WT, RT2, R6KO, and RT2; R6KO).

**Figure 2 biomedicines-09-00633-f002:**
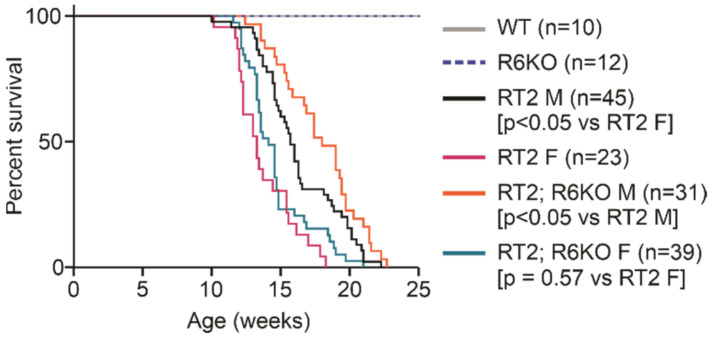
Loss of RABL6A improves survival of RT2 mice. Kaplan–Meier survival curves of overall survival for the indicated mice through 25 weeks after birth. All WT and R6KO mice were healthy and alive during this period. The Kaplan–Meier method was used to estimate the survival curves and group comparisons were made using the log rank test. *p* values for the indicated comparisons are shown; (n), number of animals in each cohort.

**Figure 3 biomedicines-09-00633-f003:**
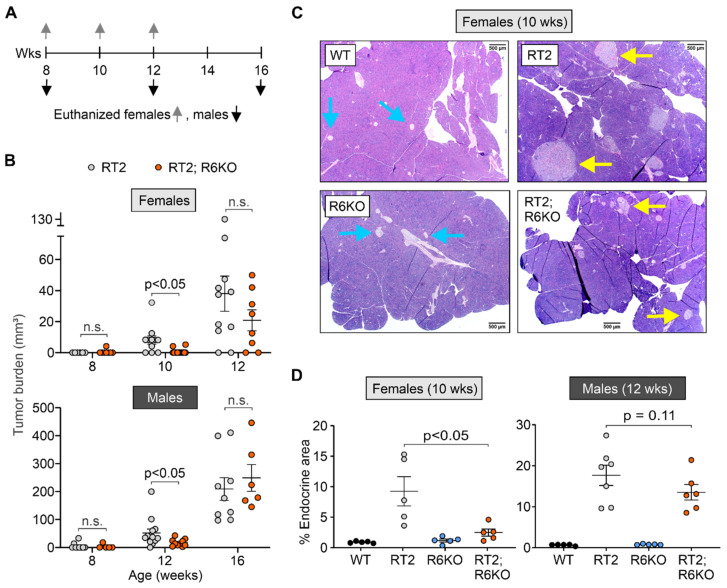
Loss of RABL6A delays tumor formation in RT2 mice. (**A**) Study timeline showing ages in weeks (wks) of female (up arrow) and male (down arrow) mice when they were euthanized for islet isolation or histopathology of the pancreas. (**B**) Tumor burden is significantly reduced in 10-week-old female and 12-week-old male RT2; R6KO mice (orange) compared to age-matched RT2 controls (gray). P values shown; n.s., not significant. N = Females: 8 weeks (RT2 = 6, RT2; R6KO = 8), 10 weeks (RT2 = 11, RT2; R6KO = 14), 12 weeks (RT2 = 11, RT2; R6KO = 8); Males: 8 weeks (RT2 = 9, RT2; R6KO = 5), 12 weeks (RT2 = 13, RT2; R6KO = 11), 16 weeks (RT2 = 9, RT2; R6KO = 6). (**C**) Representative images of H&E-stained pancreas from 10-week-old WT, RT2, R6KO, and RT2; R6KO females. Blue arrows, normal islets; yellow arrows, transformed islets. Scale bar = 500 µM. (**D**) Reduction in the % endocrine area in 10-week-old RT2: R6KO females (left, *p* < 0.05) with similar trend (*p* = 0.11) in 12-week-old RT2; R6KO males (right) versus RT2 controls. Data in A, B, and D represent the mean +/− SEM with each individual animal represented by a dot. N = Females (WT = 5, RT2 = 5, R6KO = 5, RT2; R6KO = 5); Males (WT = 5, RT2 = 7, R6KO = 5, RT2; R6KO = 6). Linear regression models were used to evaluate differences in tumor burden and beta regression models to compare percent endocrine area.

**Figure 4 biomedicines-09-00633-f004:**
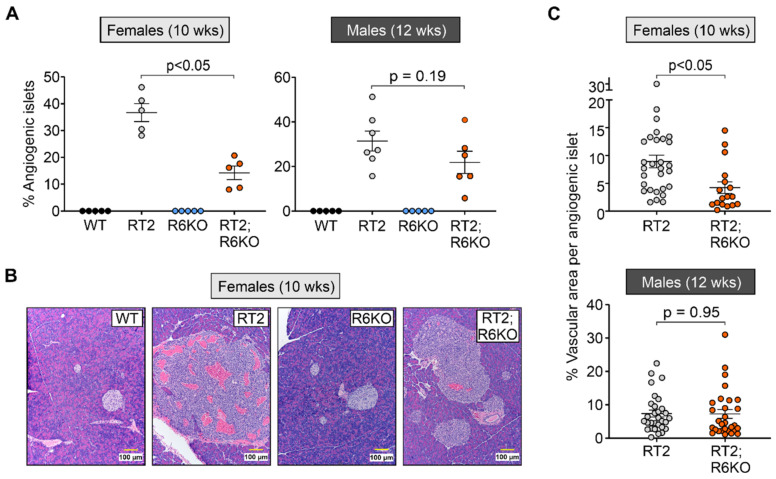
Loss of RABL6A reduces the angiogenic switch in RT2 mice. (**A**) Decreased percent angiogenic islets in 10-week-old female RT2; R6KO mice relative to RT2 controls. 12-week-old males trended similarly without statistical significance. Each dot equals data for an individual mouse pancreas. (**B**) Representative images of normal islets in 10-week-old female WT and R6KO mice versus angiogenic islets or tumors in RT2 and RT2; R6KO mice. Scale bar = 100 µm. (**C**) Reduction in percent vascular area per angiogenic islet or tumor in RT2; R6KO 10-week-old females. WT and R6KO islets lack vasculature and are not shown. Each dot represents an individual angiogenic islet or tumor. Data in A and C are reported as the mean +/− SEM. N = Females (WT = 5, RT2 = 5, R6KO = 5, RT2; R6KO = 5); Males (WT = 5, RT2 = 7, R6KO = 5, RT2; R6KO = 6). Beta regression model was used to evaluate differences in percent angiogenic islets and vascular area per angiogenic islet. Random intercepts were included to account for the correlated nature of angiogenic islets within the same mouse in the evaluation of the vascular area per angiogenic islet.

**Figure 5 biomedicines-09-00633-f005:**
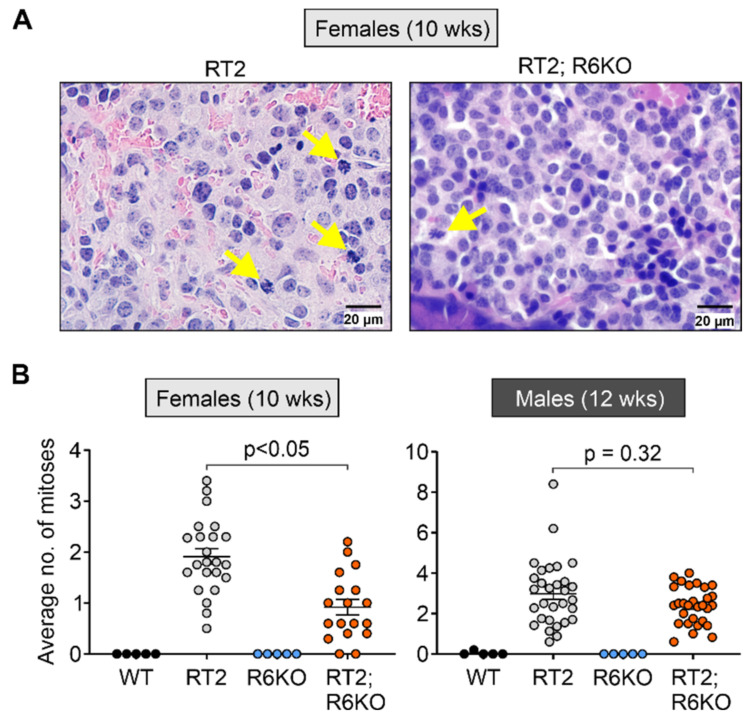
Loss of RABL6A reduces islet cell mitoses in RT2 mice. Mitoses were counted on 3–6 random fields of view within an islet at 600× magnification and their average was calculated. (**A**) Representative images of RT2 and RT2; R6KO islet sections. Arrows indicate mitoses. Scale bar = 20 µM. (**B**) The average number of mitoses (per field) is significantly reduced in the transformed islets of 10-week-old RT2; R6KO females (left) compared to age-matched RT2 females. Data are the mean +/− SEM with each dot representing an average number of mitoses per field of vision for an individual islet. N = Females (WT = 5, RT2 = 5, R6KO = 5, RT2; R6KO = 5); Males (WT = 5, RT2 = 7, R6KO = 5, RT2; R6KO = 6). Linear regression was used to compare average number of mitoses between groups.

**Figure 6 biomedicines-09-00633-f006:**
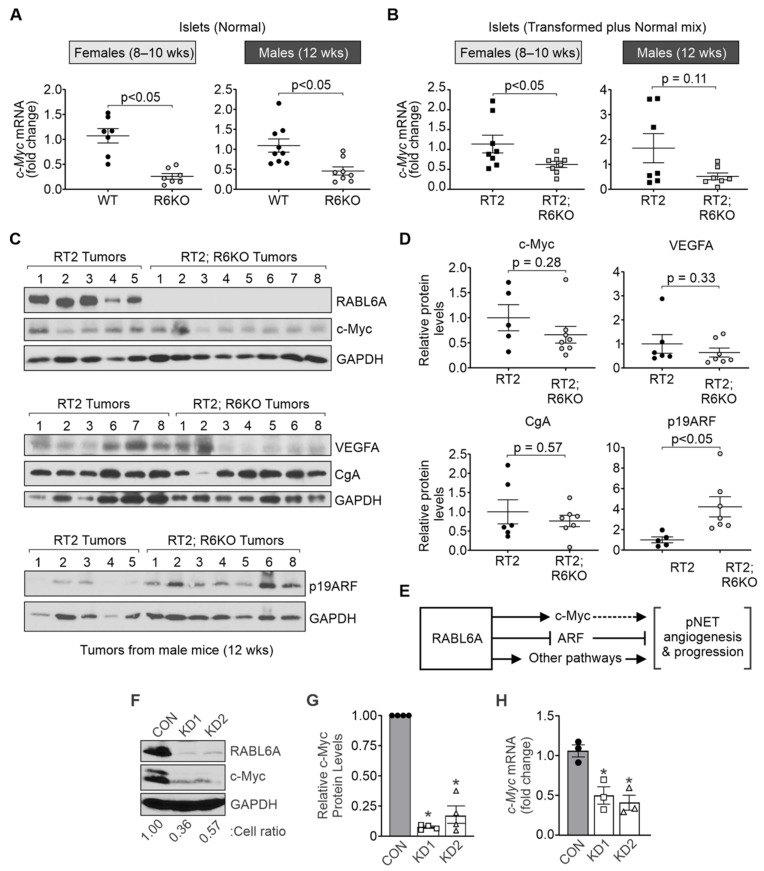
RABL6A promotes *c-Myc* and limits p19ARF expression in mouse pancreatic islets and tumors. (**A**,**B**) RNA was isolated from frozen islets (normal or a mixture of transformed plus normal), as indicated, and qRT-PCR was performed to compare the transcript levels of *c-Myc* between WT and R6KO (**A**) or RT2 and RT2; R6KO (**B**) cohorts. P values are shown. Linear regression was used to evaluate differences in the mRNA levels. Each dot represents an individual animal. N = Females (WT = 7, RT2 = 8, R6KO = 7, RT2; R6KO = 8); Males (WT = 9, RT2 = 7, R6KO = 8, RT2; R6KO = 7). (**C**) Western analysis of the indicated proteins in tumors from 12-week-old male RT2 and RT2; R6KO mice: Top-RABL6A, *c-Myc*, and GAPDH (loading control); Middle-VEGFA, Chromogranin A (CgA), and GAPDH (minus #4 and 5 of RT2 tumors and #7 of RT2; R6KO tumors; plus new #6, 7, and 8 of RT2 tumors); Bottom-p19ARF and GAPDH (minus #7 since it was used up). The different sets of blots represent separate experiments using many of the same lysates. GAPDH signals appear slightly different due to different gels and ECL exposure times. (**D**) ImageJ densitometric analysis of western data (from C) for relative levels of *c-Myc*, VEGFA, CgA and p19ARF in RT2; R6KO tumors relative to RT2 controls. Each dot represents an individual animal. N = *c-Myc* (RT2 = 5, RT2; R6KO = 8); VEGFA and CgA (RT2 = 6, RT2; R6KO = 7); p19ARF (RT2 = 5, RT2; R6KO = 7). Linear regression was used to evaluate differences in the protein levels. (**E**) Schematic summary of our findings and proposed working model for how RABL6A promotes pNET pathogenesis. (**F**–**H**) RABL6A was knocked down in human BON-1 pNET cells using two lentiviral shRNAs, KD1 and KD2. (**F**) Western blots verified RABL6A loss, which significantly reduced cell survival (as indicated by the live cell ratios of KD1 and KD2 expressing cells relative to the EV control). (**G**) Quantification of western analyses (n = 4 experiments) show *c-Myc* downregulation in RABL6A knockdown cells. (**H**) Quantification of *Myc* mRNA from qRT-PCR assays (n = 3 experiments) show it is reduced by RABL6A loss in BON-1 cells (* *p* < 0.05, Student’s t-test). Data are presented as mean +/− SEM.

**Table 1 biomedicines-09-00633-t001:** Summary of biological effects caused by RABL6A loss in RT2 mice.

Analyses		Females			Males		
		RT2	RT2; R6KO	*p*-Value	RT2	RT2; R6KO	*p*-Value
Survival (wks)	Median survival	13.3	14.1	0.57	15.7	18.0	<0.05
pNET formation	Tumor burden (mm^3^)	8.2 ± 2.7	0.7 ± 0.5	<0.05	52.0 ± 14.2	16.1 ± 3.8	<0.05
	% Endocrine area	9.3 ± 2.4	2.5 ± 0.6	<0.05	17.7 ± 2.5	13.5 ± 1.9	0.11
Islet transformation	% Angiogenic islets	36.7 ± 3.3	14.2 ± 2.5	<0.05	31.4 ± 4.5	21.9 ± 5.0	0.19
	% Vasculature area	8.9 ± 1.1	4.2 ± 1.0	<0.05	7.4 ± 1.0	7.3 ± 1.3	0.95
	Mitoses (per field of view at 600×)	1.9 ± 0.2	0.9 ± 0.2	<0.05	3.0 ± 0.3	2.4 ± 0.2	0.32
*c-Myc* expression	*c-Myc* mRNA	1.1 ± 0.2	0.6 ± 0.1	<0.05	1.7 ± 0.6	0.5 ± 0.1	0.11
	*c-Myc* protein	ND	ND	ND	1.0 ± 0.6	0.7 ± 0.2	0.28
VEGFA expression	*Vegfa* mRNA	1.2 ± 0.2	1.05 ± 0.2	0.78	1.1 ± 0.2	1.8 ± 0.3	0.05
	VEGFA protein	ND	ND	ND	1.0 ± 0.4	0.6 ± 0.2	0.33
ARF expression	*Arf* mRNA	1.9 ± 0.8	1.8 ± 0.8	0.61	1.1 ± 0.2	0.8 ± 0.4	0.26
	p19ARF protein	ND	ND	ND	1.0 ± 0.3	4.2 ± 1.0	<0.05

Data represent mean + SEM obtained from 10-week-old females and 12-week-old males (except for the survival analysis). *p*-values indicative of statistically significant differences are shown in blue. ND, not determined.

## Data Availability

Not applicable.

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
