# Peer review of "RABL6A Promotes Pancreatic Neuroendocrine Tumor Angiogenesis and Progression In Vivo"

_biomedicines, 2021, doi:10.3390/biomedicines9060633_

Round 1

Reviewer 1 Report

In this manuscript, the authors have investigated the role of RABL6A in neuroendocrine tumor. The concluded that RABL6A promotes pancreatic neuroendocrine tumor angiogenesis and progression in vivo. This is an interesting study. The authors should consider the below points to improve the manuscript.

  1. The authors should include in vitro data to further confirm the oncogenic role of RABL6A.
  2. The expression of angiogenesis markers such as VEGF should be included in Figure 4.
  3. In addition to Myc, include a couple of other biomarkers of pNET to support the conclusions. 
  4. Immunohistochemistry data is needed for RBAL6A to support their findings. 
  5. ARF and Myc immunohistochemistry data should be provided.

Reviewer 2 Report

The manuscript has investigated the role of RABL6A in pNETs. The conclusion is well supported by the data. The regulatory effect of RABL6A has been investigated in a mouse model and the results are supported by the data. I have following concerns

  1. Please cite line 41-43
  2. Please mention the groups and the number of mice in each group and experiment used for the studies in the animal model in methods section
  3. Figure 1: This Mouse models to study the in vivo role of RABL6A in pNET pathogenesis- is not a clear title, please modify
  4. Why the expression of GAPDH in western analysis of p19ARF and GAPDH # 1 and 4 is so low while this is not the case in upper panel?
  5. The results suggest that RABL6A regulate expression of c-myc, angiogenic switch, and mitosis; please support these findings by knocking down RABL6A in pNETs cell line using siRNA or knocking out RABL6A to have a direct evidence because change in expression of c-myc, angiogenic switch (VEGF, EGF, PI3K/Akt/mTOR, PKCs, Ras/MAPK) and mitosis might be due to multiple factors.

Round 2

Reviewer 1 Report

The authors have addressed all the comments appropriately and the manuscript can be accepted for publication.